# Research on the impact of coupling and coordination of rural consumption power system on the consumption of rural residents in China

Zhen Tian[1◐], Rui Wang [ID][2◐*], Yan Tan[3◐], Zhaoqin Chen[4◐]

1 Business school, Yangzhou University, Yangzhou, China, 2 Institute of Food and Strategic Reserves, Nanjing University of Finance and Economics, Nanjing, China, 3 Business School, Yangzhou University, Yangzhou, China, 4 School of Hotel and Tourism Management, Macau University of Science and Technology, Macao, China

◐ These authors contributed equally to this work.
* wangfcb37@outlook.com

## Abstract

The harmonious operation of social consumption power system can stimulate residents' higher-level consumption demands, thereby driving overall consumption growth. Based on the panel data of 30 Chinese provinces (excluding Hong Kong, Macao, Taiwan, and Tibet) from 2002 to 2022, this study integrates multiple determinants of rural residents' consumption into the conceptual framework of social consumption power. Employing a systematic approach, we construct a spatial Durbin model to empirically examine how the coupling coordination level of the rural consumption power system influences rural residents' consumption behavior. The findings reveal that the coupling coordination level of rural consumption power system exerts a statistically significant positive effect on rural residents' consumption not only within a given province but also in neighboring provinces, indicating spatial spillover effects. These spillover effects are particularly pronounced in the eastern and central regions. Therefore, in order to release the huge consumption potential in rural areas, it's vital to make good use of the coupling and coordinating role of the three subsystems of consumption subject, consumption object and consumption environment and achieve the fit and coordination between supply-side structural reform and demand-side structural reform. The research findings not only provide factual evidences and theoretical support for transforming the government's strategy of expanding rural residents' consumption but also propose practical instructions for further releasing rural residents' consumption potential in the new era.

## 1. Introduction

Due to the changes in the international division of labour as well as the international environment, China suffers problems from the external economic circulation, which

**Data availability statement:** All relevant data are within the paper and its Supporting Information files.

**Funding:** This is a part research accomplishment of the project"Research on the Evolution of Rural Residents' Consumption Behavior and the Coordinated Policy of Supply and Demand in the Past Forty Years of Reform" (No. 18BJL004))", which is supported by Office of Chinese Philosophy and Social Sciences. The funders played a role in study design, data collection and analysis, and manuscript preparation.

**Competing interests:** The authors have declared that no competing interests exist.

makes it hard to rely on the external economic impetus to achieve economic growth. In July 2020, the Chinese government proposed accelerating the establishment of a new development pattern with the domestic cycle as the focus while stimulating the domestic and international cycles to support and promote each other. The core of developing domestic circulation lies in expanding domestic demand and fully leveraging China's scale advantages and demand potential. Since the reform and opening up in China, insufficient domestic demand, especially insufficient consumption of rural residents, has restricted China's economic development. By the end of 2022, China's total population reached 1.411 billion, of which rural population of 490 million accounted for 34.8%, while the total retail sales of rural consumption goods only accounted for 13.48%. The consumption level of rural residents was much lower than that of urban residents. Therefore, it is vital to solve the problem with rural residents' consumption, explore the potential of rural consumption and promote the upgrading of consumption [1]. Meanwhile, it is quite necessary to streamline the domestic circulation to cope with fierce international competition and maintain stable and prosperous economic development [2].

Since the late 1990s, China transitioned away from shortage economy and the major restrictive factor of economic growth gradually changed from production to consumption. Coupled with the huge impact of Asian financial crisis, the central government firstly put forward the strategic policy of increasing domestic demand in February 1998 and later made it the national strategy by introducing a series of policies and measures to stimulate the consumption and expand the domestic demand. However, most policies achieved only quite limited effects because the government's purpose of expanding consumption was to increase the consumption expenditure of residents, which was not in line with the ultimate goals of meeting consumer demand and improving the quality of consumption and living standards [3]. Therefore, in order to effectively solve the problem of insufficient rural residents' consumption in China, it's not enough to only encourage rural residents to release their purchasing power and expand the consumption demand of rural society. Instead, special efforts must be taken to promote the personal development of rural residents and cultivate the consumption power of rural residents [4]. Due to the fact that consumption power is a non-linear complex comprehensive system in which three subsystems of consumption subjects, consumption objects and consumption environments coordinate with each other and function together, the entire consumption power system can run effectively and the huge consumption potential in rural areas can be released only through exploring the synergy of the system and promoting the coupling and coordinated development of three subsystems. The existing researches focus mainly on the factors that affect the consumption of rural residents such as consumption subject, consumption object and consumption environment. However, does the coupling coordination level of rural consumption power system have impact on rural residents' consumption? Is there spatial spillover effect of the coupling and coordination level of rural consumption power system on rural residents' consumption? If yes, how big is the effect? These issues are quite valuable for further study. In order to provide the reference for the government to make strategic decision on increasing

rural consumption, this paper uses the panel data of 30 Chinese provinces from 2002 to 2022 to study the impact of the coupling coordination of rural consumption power system on the consumption level of rural residents through constructing a spatial econometric model.

The subsequent parts of this paper are structured as follows. The second part puts forward the research hypothesis based on theoretical analysis. The third part applies the relevant data of the statistical yearbooks to comprehensively assess the coupling and coordination level of China's rural social consumption system through constructing the evaluation index system of the coupling and coordination levels of rural social consumption system. In the fourth part, the spatial Durbin model is applied to empirically study the impact of coupling and coordination of rural consumption power system on rural residents' consumption based on the panel data of 30 Chinese provinces from 2002 to 2022 except Hong Kong, Macao, Taiwan and Tibet. The fifth part draws conclusions and makes policy recommendations.

## 2. Theoretical analysis and research hypothesis

It's not effective enough to promote rural residents' consumption only by means of simply releasing purchasing power and increasing consumption expenditure as it may lead to low-level and low-quality repeated rural consumption without the effect of increasing rural consumption sustainably. In fact, consumption doesn't only function as being the object, motivation and purpose of production. More importantly, along with the consumers actively participating in and making consumption, more and more consumers' thinking and creativity are integrated into the consumption process, which in turn continuously improves the freedom of consumers and increases their consumption quality [5]. Therefore, in order to boost the rural residents' consumption, it's not enough to just abandon asceticism and promote consumerism. The more important is to take it as the fundamental goal to fully meet the consumption demand of rural residents and improve their life quality based on the understanding of people-oriented development. In this way, the strong driving force can be created to sustainably expand rural consumption through cultivating rural social consumption power.

### 2.1. The focus of tackling insufficient consumption: improving social consumption power

Defining consumption power as purchasing power, Adam Smith believes that purchasing power or consumption power constitutes individual income [6]. However, Marx holds the opinion that consumption power indicates the comprehensive ability of consumers to make consumption, which covers purchasing power but does not only mean it. Based on his theory, consumption power also includes the ability to clarify consumption needs, select consumption goods and get satisfaction from consumption. There are two forms of consumption power including individual consumption power and social consumption power [7]. From the macro perspective, consumption power is manifested in the form of social consumption power, which is different from absolute consumption power that refers to an ideal absolute consumption power level of the whole consumers of the society on the basis of a certain level of productivity development and social progress. Social consumption power refers to the realistic and relative level of the consumption power of the whole society, which is about the ability of all consumers to meet their consumption demand under certain conditions of production relations. This is determined by the motivation of capital accumulation, social distribution and other factors. From the micro perspective, consumption power is manifested in the form of individual consumption power, i.e., the ability of individual consumers to continuously acquire, consume and internalize consumption goods so as to meet their own needs for surviving, enjoyment and development [8].

According to Marxist political economics, production and consumption are indispensable for human beings development. If production stops, consumption will inevitably stop, and vice versa. In fact, production and consumption play equally important roles in economic growth. In the shortage economy, the market supply triggered by production investment can be effectively transformed into consumption demand, which helps to stimulate economic growth. However, along with the rapid development of productivity and the greater enrichment of products, the whole society begins to become buyer's market. If the increase in market supply triggered by the increased production investment cannot be digested and

absorbed by the market, they will become ineffective supply. This will in turn result in the imbalanced supply and demand, product backlog, production overcapacity, declining benefits, weakening growth quality as well as the negative influence on the social reproduction. Therefore, since consumption power plays a decisive role in stimulating economic growth on buyer's market [9], it is quite vital to develop consumption-driven economic growth model in the demand-constrained economy. To this end, the Central Committee of the Communist Party of China proposed in 2008 the development strategy of promoting steady and rapid economic growth by expanding domestic demand with the policy priority on building a long-term mechanism to increase consumption demand and release consumption potential so as to achieve the economic growth transformation from investment-driven mode to demand-driven one.

Consumption demand reflects the desire of consumers with payment ability to buy goods or services at a specific price level in a certain period of time, which is triggered by the combination of purchase intention and payment ability. Social consumption demand is the sum of individual consumption demand. Consumption demand is different from investment demand in the fact that the latter is a kind of intermediate demand created for production, while the former is the final one. The economic growth driven by consumption demand can effectively be accepted by the market, which can reflect the quality and efficiency of economic growth. The main reasons for the insufficient demand in China since 1998 can be explained from 2 perspectives. On the one hand, the proportion of consumption in the distribution of national income keeps being on the low side for a long time, which leads to insufficient social consumption capacity. On the other hand, people are unwilling or afraid to consume due to various reasons like social security concerns, which results in insufficient individual consumption power [10]. In fact, consumption power reflects not only the personal ability development but also the productivity advancement [11]. In addition to saving labor time, we can also develop productivity by cultivating and improving social consumption power. Therefore, in order to increase consumption demand to stimulate economic growth, individual consumption power should be cultivated through improving the quality and skills of consumers. Meanwhile, social consumption power should be enhanced as well by increasing the proportion of residents' income in the national income distribution.

## 2.2. The premise of increasing social consumption power: coupling and coordination of consumption power system

Social consumption power refers to the ability of all consumers to consume products and services in certain production relations, which indicates the relative consumption power level that the whole society can achieve. This power system consists of three basic elements: consumption subjects, consumption objects and consumption environments. Consumption subjects are the consumers that consumption power system depends on. As the material bearer of consumption power, consumption objects are something consumed, mainly including consumption materials and consumption services. Consumption environments refer to the external objective environmental factors that have some certain impact on consumption behavior, including social environment and natural environment. Consumption subjects, consumption objects and consumption environments interact and influence each other, which jointly promote the continuous improvement of residents' consumption power. Without consumption objects, consumption subjects have no way to make consumption and the former are meaningless without the latter. Consumption subjects are embedded in a certain consumption environment that affects both consumption subjects and objects in the consumption process. On the one hand, harmonious and high-quality consumption environments are conducive to enhancing consumer confidence and increasing positive consumption expectations, which thereby increases marginal propensity to consume, reduces precautionary savings, expands consumer demand and improves the level of consumption power. On the other hand, it also helps to improve the quality of consumption materials and services, improve the supply level and meet the higher level of consumption demand. Therefore, social consumption power is the complex integrated system of economy, society, nature and culture with synergistic characteristics. It is constantly formed and developed in the process of combining consumption subjects, consumption objects and consumption environments.

According to the synergy theory proposed by Hermann Haken, any system is open and composed of several subsystems. The operational effect of the whole system depends on the coupling and coordination among the subsystems [12]. 'Coupling' is originally a concept in physics, which refers to the phenomenon that two or more systems or subsystems interact with and influence each other [13]. If systems or the subsystems of a certain system can coordinate with and promote each other, it is regarded as benign coupling; otherwise it is bad. Coordination refers to the harmonious and benign correlation among two or more systems or subsystems. In order to achieve benign coupling and coordination among subsystems, it is necessary to fully make good use of system synergy and promote the interaction of material, energy and information among internal subsystems to motivate the whole system to move from disorder to order. The coupling coordination level mainly measures the level of harmony and mutual promotion among systems or subsystems, which is the main index to evaluate the level of benign coupling and measure the synergy function of the system. Since social consumption power system including consumption subjects, consumption objects and consumption environments is open, nonlinear and complex, the whole system can run effectively and the level of consumption power be steadily improved only when the system synergy is fully achieved and three subsystems are developed coordinately.

It is the premise for rural social consumption power system to operate steadily that the system composed of consumption subjects, consumption objects and consumption environments runs according to certain rules. Only when these three subsystems coordinate with each other in the same time-space serial to form the orderly, harmonious and symbiotic coupling coordination system of rural social consumption power, can the consumption demand of rural residents at the higher level be met and the consumption level of rural residents be improved. In addition, as the increase of social consumption power is generally in strict line with regional economic development, various production factors can achieve cross-regional flow beyond geographical restrictions when considered from the perspective of spatial function. The closer the geographical distance is, the higher the efficiency of cross-regional factor flow is, and the stronger the correlation between economic and social development is [14]. Therefore, the evolution of the spatial correlation and agglomeration of the coupling and coordination of rural social consumption power system is not only the outcome of the function of its own factors but also the result of the influence and promotion of the external adjacent areas [15]. The level of the coupling and coordinated development of rural consumption system in other adjacent areas may have impact on the consumption level of rural residents in the region as well.

In sum, this paper proposes the following hypothesis.

Hypothesis H1: The coupling coordination level of China's rural consumption power system has positive effect on rural residents' consumption.

Hypothesis H2: The coupling coordination level of China's rural consumption power system has spatial spillover effect on rural residents' consumption.

## 3. Comprehensive evaluation of the coupling coordination level of China's rural consumption power system

### 3.1. Evaluation method of the coupling coordination level of the rural social consumption power system

The coupling coordination level of the rural social consumption power system (D) is used to measure the benign coupling level and coordinated development level of three subsystems of consumption subjects, consumption objects and consumption environments. The specific calculation formula is as follows.

$$D = \sqrt{C \times T} \tag{2.1}$$

Among this formula, C is the coupling level of rural social consumption system and T is the coordination level of rural social consumption system.

The coupling level of rural social consumption power system C is used to describe the effect intensity among three subsystems of consumption subjects, consumption objects and consumption environments. The specific calculation formulas are as follows.

$$C = \left\{ \frac{(U_1 \times U_2 \times U_3)}{(U_1+U_2) \times (U_2+U_3) \times (U_1+U_3)} \right\}^{\frac{1}{3}}$$

(2.2)

$$U_{i=1,2,3} = \sum_{j=1}^{n} \lambda_{ij} u_{ij}, \quad \sum_{j=1}^{n} \lambda_{ij} = 1$$

(2.3)

$$U_{ij} = \frac{x_{i,j} - \min(x_{ij})}{\max(x_{ij}) - \min(x_{ij})} + A \quad U_{ij} \text{ as positive indicator}$$

(2.4)

$$U_{ij} = \frac{\max(x_{ij}) - x_{i,j}}{\max(x_{ij}) - \min(x_{ij})} + A \quad U_{ij} \text{ as negative indicator}$$

(2.5)

Among the above, $U_1$、 $U_2$ and $U_3$ are the efficacy function of three subsystems of consumption subjects, consumption objects and consumption environments respectively, which are used to evaluate the total efficacy or total contribution of three subsystems to the rural social consumption power system. $U_{ij}$ is the efficacy value of the jth index of the ith subsystem. The value range is between 0 and 1, which can usually be divided into four levels. The larger the value is, the greater the efficacy is. $U \in [0, 0.3)$ represents low level of development. $U \in [0.3, 0.5)$ represents primary level of development. $U \in [0.5, 0.7)$ represents middle level of development. $U \in [0.7, 1)$ represents the advanced level of development. Xij is the jth evaluation index of the ith subsystem of rural social consumption. The purpose of calculating the efficacy value is to make non-dimensional treatment of each index so as to eliminate the differences caused by the varieties of index and measurement units through applying the standardized method. In order to avoid the possibility that this non-dimensional treatment may lead to zero value, all the efficacy values are added with A (A=0.001). In addition, considering the fact that subjective factors may cause weight setting deviation and the index with large variation of value should be bigger in weights as it provides more information, the index weight $\lambda_{ij}$ is determined according to the modified entropy weighting method [13]. The specific steps are as follows.

Firstly, $s_{ij}$ (specific gravity of the index $U_{ij}$) is calculated as follows.

$$s_{ij} = \frac{U_{ij}}{\sum_{j=1}^{n} U_{ij}}$$

(2.6)

Secondly, $r_j$ (entropy value of the jth index) is calculated as follows.

$$\gamma_j = -\frac{1}{\ln n} \sum_{j=1}^{n} s_{\hat{r}j} \ln s_{ij}$$

(2.7)

Thirdly, $v_j$ (difference degree of the jth index) is calculated as follows.

$$V_j = 1 - r_j$$

(2.8)

Finally, $\lambda_{ij}$ (weight of the jth index) is calculated as follows.

$$\lambda_{ij} = \frac{V_j}{\sum_{j=1}^n V_j}$$

(2.9)

The coordination level of rural social consumption power system T is used to describe the benign level of the coupling relationship among three subsystems of consumption subjects, consumption objects and consumption environments so as to reflect whether the subsystems promote each other at a high level or restrict each other at a low level. The specific calculation formula is as follows.

$$T = \alpha U_1 + \beta U_2 + \gamma U_3$$

(2.10)

Among the formula, α, β and γ are the weights of three subsystems of consumption subjects, consumption objects and consumption environments respectively. As three subsystems are roughly equal to rural social consumption power system in the effect and importance, α, β and γ are uniformly assigned with the value of 1/3.

The value range of the coupling coordination level of rural social consumption power system is between 0 and 1. The larger the value is, the higher the level of coordinated development of the system is. When the coupling coordination level of rural social consumption power system D∈[0, 0.3), it reflects that rural social consumption power system is generally in the low-level development stage featured with imbalance and disorder, but it may be accumulating new potential energy towards orderly development stage. When the coupling coordination level of rural social consumption power system D∈[0.3, 0.5), it reflects that rural social consumption power system is in the primary stage featured with mild imbalance and begins to gradually enter into the track of orderly development. When the coupling coordination level of rural social consumption power system D∈[0.5, 0.7), it reflects that rural social consumption power system begins to develop towards the coordinated and orderly direction and gradually enters into the middle development stage of running-in and adaptation. When the coupling coordination level of rural social consumption power system D∈[0.7, 1), it reflects that the positive accumulation effect of harmony and mutual promotion among the subsystems within rural social consumption power system is more obvious and the system has entered into the coordinated and orderly advanced development stage.

## 3.2. Construction of evaluation index system of coupling coordination level of the rural social consumption power system

Only when the orderly, harmonious and symbiotic rural social consumption power system is formed with the consumption subject, consumption object and consumption environment interacting with and influencing each other, can the higher-level rural residents' consumption demand be met and rural residents' consumption be promoted. Although there exist rich research findings about the influence of consumption subject, consumption object and consumption environment on rural residents' consumption, little research has been made on the influence of the coupling and coordination of rural social consumption system composed of consumption subject, consumption object and consumption environment on rural residents' consumption. Following the principles that the index system should be constructed scientifically, systematically and feasibly, this paper selects the indexes widely used by the scholars after sorting out and analyzing the above-mentioned literature through the frequency statistics method. Taking the availability of data into account, the paper selects and optimizes these indicators, and finally constructs the evaluation index system of the coupling coordination level of rural social consumption power system (see Table 1). Among the indexes, the consumption subject subsystems and the consumption environment subsystems include 10 basic indexes, and the consumption object subsystems include 9 basic indexes. This paper takes 30 Chinese provinces as the research objects and selects relevant data from 2002 to 2022 for empirical analysis, which are

**Table 1. The coupling and coordination evaluation index system of rural social consumption power system and the weight of each index.**

| Subsystem | Index | Indicator description | Weight |
|---|---|---|---|
| Consumption subject | Rural population | Unit of measurement: million people | 0.080 |
| | Per capita disposable income of rural residents | Unit of measurement: yuan | 0.129 |
| | Per capita disposable income growth rate of rural residents | (Current income – Previous period income)/ Previous period income | 0.081 |
| | Income gap between urban and rural residents | Per capita disposable income of urban residents – Per capita disposable income | 0.086 |
| | Rural household assets | Per capita property income of rural residents | 0.099 |
| | Social security level of rural residents | Per capita transfer income of rural residents | 0.116 |
| | Liquidity constraints | (Cash income – Cash expenditure)/ Cash income | 0.093 |
| | Consumption habits of rural residents | Per capita consumption expenditure of rural residents in the previous period | 0.098 |
| | Population dependency ratio | Non-working-age population/ Working population | 0.040 |
| | Educational level of rural residents | Proportion of rural population with high school education or above | 0.168 |
| Consumption object | Agricultural productivity | Agricultural added value/ Agricultural employment | 0.106 |
| | Degree of industrialization | Proportion of the secondary industry output in GDP | 0.155 |
| | Degree of opening to the outside world | Proportion of import and export volume in GDP | 0.096 |
| | Scale of circulation industry | Total value of wholesale and retail industry/ total value of tertiary industry | 0.140 |
| | Consumption of energy available | Unit of measurement: ten thousand tons of standard coal | 0.112 |
| | Total output value of agriculture, forestry, animal husbandry and fishery | Unit of measurement: billion yuan | 0.104 |
| | Catering business area at the end of the year | Unit of measurement: square meter | 0.041 |
| | Total retail sales of consumption goods | Unit of measurement: billion yuan | 0.128 |
| | Freight turnover | Unit of measurement: million tons | 0.048 |
| Consumption environment | Urbanization rate | Urban population/ Total population | 0.119 |
| | Density of rural road network | Highway mileage/ Urban area | 0.071 |
| | Rural internet penetration rate | Proportion of rural internet access users in the total rural households | 0.085 |
| | Rural medical and health conditions | Number of beds in rural health institutions | 0.099 |
| | Rural education conditions | Proportion of rural full-time teachers in the total rural employees | 0.177 |
| | Rural security situation | Number of cases handled per 10,000 people in rural areas | 0.062 |
| | Financial support for agriculture | Local financial expenditure on agriculture, forestry and water affairs | 0.134 |
| | Passenger transport transit volume | Unit of measurement: million people | 0.076 |
| | Basic medical insurance coverage rate for rural residents | Proportion of rural residents buying medical insurance | 0.171 |
| | Pension insurance coverage rate for rural residents | Proportion of rural residents buying pension insurance | 0.084 |

referenced from China Statistical Yearbook, China Rural Statistical Yearbook, China Population and Employment Statistical Yearbook, China Energy Statistical Yearbook, China Education Statistical Yearbook, China Urban and Rural Construction Statistical Yearbook and statistical yearbooks of 30 provinces (municipalities and autonomous regions). In order to eliminate the influence of dimension and data levels, the original data are standardized through Formula 2.4 and 2.5. Then, the weight of each index of three subsystems of rural social consumption power from 2002 to 2022 is calculated by using the modified entropy weighting method (Formula 2.6~Formula 2.9). Finally, the mean value of these 21-year index values is taken as the final weight value of the index weight $\lambda_{ij}$ (see Table 1) to eliminate the difference in weight calculation for different years and the fluctuation of weight value caused by special events in some years.

### 3.3. Descriptive statistical analysis of coupling coordination level of rural consumption power system

The following analysis can be made based on the descriptive statistical analysis of the coupling coordination level of rural consumption power system in 30 Chinese provinces from 2002 to 2022 (see Table 2). As for the maximum value, the coupling coordination level of rural consumption power system increases from 0.738 in 2002 to 0.866 in 2022, and the overall coupling coordination level demonstrates an upward trend. The areas with high coupling coordination level of rural consumption power system are mainly for those economically developed areas such as Beijing, Shanghai, Guangdong, Jiangsu and Zhejiang with the provinces of the highest level of coupling coordination level reaching the high-quality coordination level. As for the minimum value, the coupling coordination level of rural consumption power system increases from 0.382 in 2002 to 0.556 in 2022. The coupling coordination level of rural consumption power system in the underdeveloped areas also demonstrates an overall upward trend, which is mainly for Guizhou, Gansu, Ningxia and other central and western provinces of China. The coupling coordination level of rural consumption power system in some provinces is in a state of mild imbalance before 2005, after which it gradually evolves to the verge of imbalance. Since 2016, the coupling coordination level of rural consumption power system in all provinces of China has been beyond the barely coordinated level. As for the range value, the coupling coordination level of rural consumption power system in each year is about 0.35 with the trend decreasing, which indicates that there is polarization phenomenon in the coupling coordination level of rural consumption power system in different provinces of China, but the gap is gradually narrowing. As for the average value, the coupling coordination level of rural consumption power system has been at the barely coordinated level since 2002 and it is almost close to the middle development stage of running-in and adaptation in 2022 with the steady upward trend. As for standard deviation, the coupling coordination level of rural consumption power system in 30

**Table 2. Descriptive statistical analysis of the coupling coordination level of rural consumption power system in Chinese provinces from 2002 to 2022.**

| Year | Max | Min | Average | Median | Sd |
|------|-----|-----|---------|--------|-----|
| 2002 | 0.738 | 0.382 | 0.559 | 0.589 | 0.132 |
| 2003 | 0.764 | 0.389 | 0.565 | 0.596 | 0.128 |
| 2004 | 0.760 | 0.399 | 0.579 | 0.610 | 0.126 |
| 2005 | 0.757 | 0.406 | 0.577 | 0.613 | 0.125 |
| 2006 | 0.776 | 0.408 | 0.586 | 0.627 | 0.128 |
| 2007 | 0.788 | 0.422 | 0.594 | 0.627 | 0.127 |
| 2008 | 0.791 | 0.435 | 0.601 | 0.642 | 0.125 |
| 2009 | 0.795 | 0.443 | 0.605 | 0.649 | 0.124 |
| 2010 | 0.798 | 0.458 | 0.621 | 0.672 | 0.124 |
| 2011 | 0.801 | 0.470 | 0.629 | 0.675 | 0.122 |
| 2012 | 0.804 | 0.481 | 0.638 | 0.686 | 0.120 |
| 2013 | 0.811 | 0.480 | 0.642 | 0.688 | 0.120 |
| 2014 | 0.814 | 0.487 | 0.652 | 0.698 | 0.118 |
| 2015 | 0.818 | 0.489 | 0.655 | 0.696 | 0.119 |
| 2016 | 0.824 | 0.502 | 0.662 | 0.701 | 0.117 |
| 2017 | 0.842 | 0.516 | 0.670 | 0.710 | 0.114 |
| 2018 | 0.847 | 0.520 | 0.680 | 0.719 | 0.113 |
| 2019 | 0.852 | 0.519 | 0.687 | 0.724 | 0.114 |
| 2020 | 0.855 | 0.538 | 0.694 | 0.739 | 0.113 |
| 2021 | 0.860 | 0.547 | 0.702 | 0.748 | 0.112 |
| 2022 | 0.866 | 0.556 | 0.710 | 0.755 | 0.111 |

Source: Calculated by the author.

Chinese provinces from 2002 to 2022 is between 0.111 and 0.132 with the trend gradually decreasing coupled with small fluctuations, which indicates that the dispersion degree of the coupling coordination level of rural consumption power system is not high with the trend of getting smaller. As time goes by, the coupling coordination level of rural consumption power system in each province begins to converge gradually.

## 4. Empirical study on the impact of coupling coordination level of rural consumption power system on rural residents' consumption in China

### 4.1. Model construction

According to Waldo Tobler's spatial correlation law, there is inevitable connection between things, which is related to distance as well. Generally, the closer the distance is, the greater the correlation is. The farther the distance is, the greater the dissimilarity is. From spatial dimension, all kinds of production factors can effectively achieve cross-regional flow beyond regional restrictions with higher efficiency of factor flow between closer regions. Therefore, the regions with similar geographical locations usually have stronger correlation in economic development. There is obvious spatial heterogeneity in the coupling coordination level of rural consumption power system in different regions of China due to the high consistency between rural consumption power system and local economic development. The impact of rural consumption power system on the consumption level of rural residents is not only limited by the coupling coordination level of three subsystems of consumption subjects, consumption objects and consumption environments, but also possibly affected by spatial interaction. That is to say, the consumption level of rural residents in one region may be influenced by the coupling coordination development level of rural consumption power system in other adjacent areas. According to the spatial econometrics developed in the 1970s, the correlation caused by the geographical proximity of economic variables is highly valued. Therefore, this paper chooses to construct spatial econometric model to empirically study the impact of the coupling coordination level of rural consumption power system on the consumption level of rural residents.

The main purpose of constructing spatial econometric model is to establish spatial weight matrix to measure the spatial correlation of economic variables in different regions. It can be found from current research findings that there are four common ways to establish the spatial weight matrix including spatial adjacency matrix based on geographical proximity, geographical distance matrix based on geographical location, economic distance matrix based on both geographical and economic distance, and nested matrix based on geographical and economic attributes [2]. In this research, geographical distance matrix cannot reflect the sample data well due to the lack of representativeness of some provinces. The economic distance matrix is not stable because economic development level changes with time. The nested matrix may negatively affect the estimation of the spatial econometric model due to the large probability of misuse in the application of this matrix [16]. Considering the fact that Chinese provinces have been clearly divided geographically in recent years and the geographical location of adjacent regions of a certain province are fixed, this paper applies the spatial adjacency matrix to obtain the spatial weight to reflect the geographical characteristics of each province in China.

$$Y = \rho W_N Y + \alpha X + \varepsilon \tag{3.1}$$

The spatial econometric model mainly includes two types of Spatial Autoregressive Model (SAR) and Spatial Error Model (SEM). SAR model assumes that the value of the dependent variable is affected by adjacent samples with spatial correlation, while SEM model assumes that the dependent variable has spatial correlation based on its own characteristics. The specific formula of SAR model is as follows.

$$Y = \rho W_N Y + \alpha X + \varepsilon \tag{3.2}$$

In the formula, Y represents the explained variable. X represents the explanatory variable or control variable. $W_N$ represents the spatial weight matrix. ρ is the spatial autoregressive coefficient, indicating the mutual influence between

adjacent regions. α is the regression coefficient, indicating the influence of adjacent areas on the explained variables in the region. ε is the random disturbance term. The specific formula of SEM model is as follows.

$$Y = \alpha X + \mu \tag{3.3}$$

$$\mu = \theta W_N + \varepsilon \tag{3.4}$$

In the formula, Y represents the explained variable. X represents the explanatory variable or control variable. $W_N$ is the spatial weight matrix. θ is the spatial autoregressive coefficient, indicating the mutual influence between adjacent regions. ε is the random disturbance term.

When the lag terms of endogenous and exogenous variables are introduced into SAR model and SEM model, Spatial Durbin model (SDM) can be created as follows:

$$Y = \rho W_N Y + \alpha_1 X + \alpha_2 W_N X + \varepsilon \tag{3.5}$$

In the formula, Y represents the explained variable. X represents explanatory variable or control variable. $W_N$ is the spatial weight matrix. ρ is the spatial autoregressive coefficient, indicating the mutual influence between adjacent regions. $\alpha_2$ is the regression coefficient, indicating the influence of adjacent areas on the explained variables in a certain region. ε is the random disturbance term. The specific model that is selected to make empirical analysis depends on the test and the significant result.

## 4.2. Variable selection and descriptive statistics

**4.2.1. Explained variables.** Rural residents' consumption (CON): This index is measured by per capita consumption expenditure of rural residents. In order to eliminate the influence of price factors, this paper takes 2002 as the base year and uses the rural consumption price index of each province to adjust the explained variable. Meanwhile, as per capita consumption expenditure of rural residents is relatively big, the logarithmic form is adopted.

**4.2.2. Core explanatory variables.** The coupling coordination level of rural consumption power system (DCC): This index is measured by the coupling coordination level of rural consumption power system in the corresponding years of each province as calculated above.

**4.2.3. Control variable.** Income of rural residents (INC): Classical theories about consumption function and existing researches demonstrate that income is the key factor affecting the consumption level with the income increase positively influencing the improvement of consumption level [17]. This index is measured by disposable income of rural residents. As its value is relatively big, the logarithmic form is adopted. The elderly dependency ratio (ODR) and children dependency ratio (CDR): According to the life cycle hypothesis proposed by American economists F. Modigliani & R. Brumberg, changes in the age structure of the population affect the total amount and structure of household consumption through influencing the consumption decisions of individuals [18]. This paper uses the elderly dependency ratio (ODR) and the children dependency ratio (CDR) to measure the age structure of rural population. The increase of the elderly dependency ratio can enhance the motivation of precautionary savings and thus suppress household consumption to a certain extent, while the increase in children dependency ratio can directly raise the consumption expenditure on those things like life and education. However, some researchers suggest that general tradition of saving for education and marriage of the under-aged children in rural households in China may increase the household motivation to make precautionary savings and thus reduce immediate consumption [19]. Educational level of rural residents (EDU): The rapid development of new economy has led to big changes in consumption patterns and also put forward higher requirements for rural residents to have more consumption ability that is based on the improvement of personal scientific and cultural level. In addition, the

improvement of the scientific and cultural level of rural residents can help to increase the consumption of rural residents by increasing their human capital to generate income effects [20]. Therefore, this paper measures the educational level of rural residents with the index of the number of students in ordinary high schools per 10,000 people in rural areas. As the value is relatively big, the logarithmic form is adopted. Rural medical level (RML): The level of rural medical care can directly affect the welfare of rural residents. Improved rural medical level can effectively reduce the precautionary savings of rural residents and increase consumption intention and ability [21], thus increasing the consumption of rural residents [22]. Therefore, this paper measures the level of rural medical care with the number of technical personnel in health institutions per 10,000 people in rural areas. As its value is relatively big, its logarithmic form is adopted. Rural price level (RPL): The dynamic changes and interaction of price and income can impact the actual consumption of residents. If the growth rate of price level is higher than that of income, the actual consumption level of residents will decline, and vice versa [23]. This paper uses the rural consumption price index to measure the rural price level. Level of financial support for agriculture (FSA): Fiscal expenditure for supporting agriculture can not only indirectly stimulate rural residents' consumption through income effect but also effectively improve rural consumption environments, enhance rural consumption potential and increase rural residents' consumption demand [24]. This paper measures the level of financial support for agriculture with local expenditure on agriculture, forestry and water. As its value is relatively big, the logarithmic form is adopted. Level of urbanization (URB): The development of urbanization not only promotes the transfer of rural surplus labor force and the continuous and stable growth of rural residents' income [25], but also enhances the citizenization of rural residents, consumption psychology, consumption habits and consumption patterns, which improves the total consumption ability. It also promotes market development, improves the rural infrastructure and upgrades the rural consumption environment and living space, thus enhancing the consumption level of rural residents and stimulating the growth of rural consumption demand [26]. This paper measures the level of urbanization with the proportion of urban population in the total of each province.

This paper empirically studies the impact of the coupling coordination level of rural consumption power system on the consumption level of rural residents based on the observation of 30 Chinese provinces from 2002 to 2022. The data of the core explanatory variables are derived from the coupling coordination level of rural consumption power system in the corresponding years of each province as calculated above. Other data are from China Statistical Yearbook, China Population and Employment Statistical Yearbook, China Rural Statistical Yearbook and the statistical yearbooks of the covering provinces (municipalities and autonomous regions) from 2002 to 2022. All economic indicators have been deflated according to the corresponding index of the year 2002. Meanwhile, in order to minimize the influence of heteroscedasticity and take the same dimensions, the corresponding data are processed logarithmically. The specific descriptive statistics of all indicators are exhibited in Table 3.

## 4.3. Spatial autocorrelation test

According to the spatial adjacency matrix based on geographical adjacency made in the previous part, this paper selects Global Moran's Indicator to test the global spatial autocorrelation of China's rural residents' consumption level and the coupling coordination level of China's rural consumption power system from 2002 to 2022 through Stata16 software. The results demonstrate that the coupling coordination level of China's rural consumption power system and Chinese rural residents' consumption level have significant spatial positive correlation with obvious spatial dependence and regional correlation(see Table 4). Therefore, it is necessary to consider the spatial spillover effect in the study of the impact of coupling coordination level of rural consumption power system on rural residents' consumption, which proves it appropriate to choose the spatial econometric model.

In order to further examine the spatial correlation of variables, this paper draws Moran scatter plot of the coupling coordination level of China's rural consumption power system and the consumption level of rural residents. The results demonstrate that the observed values of most provinces fall in the first and the third quadrants, which means that these

**Table 3. Concrete meaning of indicators and descriptive statistical analysis.**

| Evaluating indicator | Concrete meaning | Unit | Mean | Sd | Min | Max |
|---|---|---|---|---|---|---|
| Rural residents' consumption | Per capital consumption expenditure of rural residents | Yuan | 8.688 | 0.789 | 6.950 | 10.403 |
| Coupling coordination level of rural consumption power system | Coupling coordination level of rural consumption power system | – | 0.638 | 0.132 | 0.382 | 0.936 |
| Rural residents' income | Per capita disposable income of rural residents | Yuan | 8.913 | 0.756 | 7.288 | 10.598 |
| Children dependency ratio | Proportion of population aged 0–14 years old in that aged 15–64 years old | % | 24.5 | 7.2 | 9.6 | 44.7 |
| The elderly dependency ratio | Proportion of population aged 65 and above in the population aged 15–64 | % | 15.1 | 0.8 | 6.7 | 55.8 |
| Educational level of rural residents | Number of ordinary high school students per 10,000 rural population | – | 4.894 | 0.502 | 3.449 | 5.989 |
| Rural medical level | Number of technical personnel in health institutions per 10,000 people in rural areas | – | 3.944 | 0.357 | 2.978 | 4.905 |
| Rural price level | Rural consumption price index | – | 1.025 | 0.020 | 0.975 | 1.123 |
| Level of financial support for agriculture | Local expenditure on agriculture, forestry and water affairs | Billion Yuan | 5.372 | 1.187 | 2.553 | 7.214 |
| Urbanization level | Proportion of urban population in the total population of the province. | % | 48.4 | 16.6 | 19.8 | 94.1 |

**Table 4. Global Moran's I of the coupling coordination level of Chinese rural consumption power system and the consumption level of rural residents from 2002 to 2022.**

| Year | Coupling coordination level of rural consumption power system | | | Consumption level of rural residents | | |
|---|---|---|---|---|---|---|
| | Moran's I | P-value | Z-value | Moran's I | P-value | Z-value |
| 2002 | 0.298 | 0.001 | 3.404 | 0.342 | 0.000 | 4.061 |
| 2003 | 0.329 | 0.001 | 3.725 | 0.340 | 0.000 | 4.046 |
| 2004 | 0.331 | 0.001 | 3.748 | 0.356 | 0.000 | 4.240 |
| 2005 | 0.353 | 0.001 | 3.964 | 0.362 | 0.000 | 4.292 |
| 2006 | 0.369 | 0.001 | 4.132 | 0.386 | 0.000 | 4.556 |
| 2007 | 0.372 | 0.001 | 4.168 | 0.384 | 0.000 | 4.541 |
| 2008 | 0.372 | 0.001 | 4.159 | 0.359 | 0.000 | 4.211 |
| 2009 | 0.391 | 0.001 | 4.360 | 0.326 | 0.000 | 3.856 |
| 2010 | 0.383 | 0.001 | 4.277 | 0.351 | 0.000 | 4.113 |
| 2011 | 0.387 | 0.001 | 4.316 | 0.377 | 0.000 | 4.351 |
| 2012 | 0.384 | 0.001 | 4.291 | 0.393 | 0.000 | 4.501 |
| 2013 | 0.378 | 0.001 | 4.230 | 0.364 | 0.000 | 4.183 |
| 2014 | 0.366 | 0.001 | 4.109 | 0.301 | 0.000 | 3.515 |
| 2015 | 0.360 | 0.001 | 4.051 | 0.339 | 0.000 | 3.901 |
| 2016 | 0.374 | 0.001 | 4.18 | 0.429 | 0.000 | 3.839 |
| 2017 | 0.375 | 0.001 | 4.193 | 0.443 | 0.000 | 3.972 |
| 2018 | 0.377 | 0.001 | 4.207 | 0.4577 | 0.000 | 4.106 |
| 2019 | 0.378 | 0.001 | 4.220 | 0.471 | 0.000 | 4.239 |
| 2020 | 0.339 | 0.001 | 3.830 | 0.282 | 0.001 | 3.295 |
| 2021 | 0.335 | 0.001 | 3.801 | 0.308 | 0.000 | 3.557 |
| 2022 | 0.355 | 0.001 | 4.008 | 0.293 | 0.001 | 3.395 |

provinces have strong positive promotional effects in local space with the feature of typical spatial aggregation. This finding is in line with the test results of Global Moran's indicators as analysed above. In addition, the observed values from 2002 to 2022 that fall within the scope of the first and the third quadrants gradually increase, which indicates that the spatial aggregation of the coupling coordination level of China's rural consumption power system and the rural residents' consumption level in local areas are significantly enhanced. In sum, the coupling coordination level of China's rural consumption power system and the consumption level of rural residents both have certain spatial dependence. As the ordinary panel model has big regression error, the spatial panel model should be selected.

## 4.4. Spatial econometric model selection

Firstly, spatial correlation tests including LM-Lag, RLM-Lag, LM-Error and RLM-Error tests (see table 5) are conducted on ordinary panel regression to determine the type of spatial effect and select the model form. According to LM test results, the statistics of LM-Lag, RLM-Lag and LM-Error tests are all significant at the 1% level, while the statistics of RLM-Error test are significant at the 5% level. All the four tests deny the null hypothesis, which indicates that it is more reasonable to choose the spatial Durbin model (SDM) because the sample has dual effects of spatial error autocorrelation and spatial lag. Secondly, the Hausman index of SDM model in Hausman test is 22.17 and passes the test at the 1% significance level, which indicates that it is more appropriate to choose the fixed effect model. Finally, LR test is carried out on the spatial Durbin fixed effect model with the result that the index value of LR test denies the null hypothesis at the significance level of 1%. Meanwhile, the Wald test is conducted to test the robustness with the result of 16.07, which is also valid at the significance level of 1%. It indicates that the spatial Durbin model (SDM) cannot be degraded into Spatial Autoregressive Model (SAR). In sum, this paper chooses the spatial Durbin fixed effect model to study the impact of the coupling coordination level of China's rural consumption power system on rural residents' consumption.

## 4.5. Spatial econometric model selection

Based on the panel data of 30 Chinese provinces from 2002 to 2022, this research applies spatial Dubin individual fixed effect model, time fixed effect model and two-way fixed effect model to test the effect of the coupling coordination level of rural consumption power system on rural residents' consumption by using Stata16 software (see Table 6). The regression coefficient of explanatory variables indicates that two-way fixed effect model has the highest level of significance and the regression effect of the model is relatively the best. Therefore, this paper chooses two-way fixed effect model to make explanation.

According to the regression results of the model, the total effect coefficient of the coupling coordination level of China's rural consumption power system on the consumption level of rural residents is 0.459. The hypothesis test is verified at the significance level of 1%. It shows that the coupling coordination level of rural consumption power system plays significant role in promoting the improvement of rural residents' consumption level, which verifies the hypothesis H1. The spatial lag coefficient of the coupling coordinated level of China's rural consumption power system is 0.251 with the original hypothesis denied at the significance level of 5%. This indicates that the coupling and coordinated development level of rural

**Table 5. Spatial correlation LM test results.**

| Test | Statistic | P -Value |
|------|-----------|----------|
| LM-Lag | 14.340*** | 0.000 |
| RLM-Lag | 16.119*** | 0.000 |
| LM-Error | 21.475*** | 0.001 |
| RLM-Error | 39.508** | 0.067 |

Note: * * *, * * and * represent being significant at the level of 1%, 5%, and 10% respectively.

**Table 6. The spatial effect of the coupling coordination level of rural consumption power system on the consumption level of rural residents.**

| Variable | Individual fixed effect | Time-fixed effect | Two-way fixed effects |
|---|---|---|---|
| DCC | 0.301 | 0.243 | 0.459*** |
|  | (2.98) | (2.11) | (2.52) |
| INC | 0.304*** | 0.252*** | 0.325*** |
|  | (0.024) | (0.026) | (2.41) |
| ODR | −0.093 | −0.318 | −0.001*** |
|  | (0.260) | (0.038) | (−5.07) |
| CDR | 0.032 | 0.119 | 0.466* |
|  | (0.311) | (0.139) | (2.23) |
| EDU | 0.109*** | 0.039 | 0.004* |
|  | (0.043) | (0.031) | (0.114) |
| RML | 0.039 | −0.095*** | −0.031*** |
|  | (0.030) | (0.028) | (−4.49) |
| RPL | −0.046 | −0.287 | −0.031* |
|  | (0.116) | (0.067) | (−1.82) |
| FSA | 0.027 | 0.108 | 0.088*** |
|  | (0.0688) | (0.0433) | (0.042) |
| URB | 0.224* | 0.304*** | 0.523*** |
|  | (0.115) | (0.115) | (4.96) |
| Spatial rho | −0.017 | −0.715*** | 0.251** |
|  | (1.86) | (4.92) | (4.68) |
| Variance sigma2_e | 0.132*** | 0.235*** | 0.235** |
|  | (1.11) | (2.96) | (2.96) |
| N | 630 | 630 | 630 |
| R² | 0.223 | 0.527 | 0.923 |

Note: * * *, * * and * represent being significant at the level of 1%, 5% and 10% respectively.

consumption power system has significant spatial effect, which means that the coupling and coordinated development level of rural consumption power system in one province has significant promoting effect on the consumption level of the rural residents in other provinces. This verifies Hypothesis H2 in turn.

Although spatial Durbin model can be used to explain the spatial effect of the coupling coordination level of rural consumption power system on the consumption level of rural residents, the parameter estimation results have errors without being able to fully reflect the direct effect and spatial spillover effect because the spatial lag term of the independent variable ignores the feedback between adjacent regions. Therefore, this paper uses the partial differential method proposed by Le Sage et al. to decompose the effects of the variables on the consumption level of rural residents into direct effect, indirect effect and total effect [27]. It can be seen from Table 7 that the direct effect, indirect effect and total effect of the coupling coordination level of China's rural consumption power system on the consumption level of rural residents are all positive and significant in the test at the 1% level, which indicates that the coupling coordination level of rural consumption power system not only effectively promotes the improvement of the consumption level of rural residents in the province but also has obvious spatial spillover effect. If the influence of spatial factors is ignored in model construction, the promotional effect of the coupling coordination level of rural consumption power system on the consumption level of rural residents will be underestimated.

It can be seen from the regression results of the model that the direct effect, indirect effect and total effect of rural residents' income level on rural residents' consumption level are all positive and significant at the level of 1%, which indicates

**Table 7. Decomposition of the spatial effect of the influence of the coupling coordination level of rural consumption power system on rural residents' consumption level.**

| Variable | Direct effect | Indirect effect | Gross effect |
|---|---|---|---|
| DCC | 0.322** | 0.252** | 0.574** |
| | (2.91) | (3.01) | (5.64) |
| INC | 0.381*** | 0.362*** | 0.743*** |
| | (3.15) | (4.07) | (8.09) |
| CDR | 0.415*** | 0.020 | 0.435*** |
| | (3.27) | (1.21) | (3.02) |
| ODR | −0.006*** | −0.0001 | −0.006*** |
| | (−3.41) | (−1.28) | (−3.16) |
| EDU | 0.032*** | 0.272** | 0.304*** |
| | (3.11) | (5.54) | (3.51) |
| RML | −0.113*** | −0.246** | −0.359*** |
| | (3.01) | (−3.89) | (−5.10) |
| RPL | −0.251*** | −0.252 | −0.503*** |
| | (2.62) | (2.42) | (6.77) |
| FSA | 0.009*** | 0.001*** | 0.010*** |
| | (2.60) | (2.86) | (5.44) |
| URB | 0.110*** | 0.045 | 0.155*** |
| | (3.09) | (1.30) | (2.93) |

Note: $***$, $**$ and $*$ represent being significant at the level of 1%, 5% and 10% respectively.

that the increase of rural residents' income level not only directly promotes rural residents' consumption level but also has positive spatial spillover effect. The direct effect and total effect of children's dependency ratio on the consumption level of rural residents are significantly positive as well, while the indirect effect is not significant. It shows that the consumption expenditures on education and heath increase along with higher children's dependency ratio because rural households put more emphasis on children's education and health [28], which inevitably improves the consumption level of rural residents. But it has no spatial spillover effect with little influence on the consumption level of rural residents in the adjacent areas. The direct effect and total effect of the elderly dependency ratio on the consumption level of rural residents are significantly negative, which is mainly due to the fact that most old people are not only low in income but also thrifty in consumption. In addition, the widespread bequest motivation suppresses the improvement of the consumption level of rural residents [29]. However, the indirect effect of the elderly dependency ratio on the consumption level of rural residents is not significant. The direct effect, indirect effect and total effect of rural residents' educational level on rural residents' consumption level are all positive and significant, which indicates that the improvement of rural residents' educational level can not only enhance their human capital and directly improve their consumption level through income effect, but also promote the consumption level of the surrounding rural residents through indirectly improving their consumption cognition, attitude, knowledge and skills [30]. In sum, there is positive spatial spillover effect of this variable. In view of the problems of rural medical system such as the underdeveloped facilities, unreasonable structure, insufficient human resources and lagging team construction, the direct effect, indirect effect and total effect of rural medical level on rural residents' consumption level are negative and significant at least at the level of 5%, which indicates that rural medical level not only suppresses the improvement of rural residents' consumption level but also has negative spatial spillover effect. The direct effect and total effect of rural price level on the rural residents' consumption level are negative with insignificant indirect effect. This indicates that the growth rate of rural residents' income level is less than that of price growth, which

suppresses the consumption of rural residents. However, it does not affect the consumption level of rural residents in the adjacent areas. The direct effect, indirect effect and total effect of the financial support for agriculture on the consumption level of rural residents are positive and significant at the level of 1%, which indicates that the increase of financial support for agriculture not only helps improve rural infrastructure construction and create good rural consumption environment but also has significant spatial spillover effect [31]. Thus the consumption level of rural residents in both local and adjacent areas can be promoted. The direct effect and total effect of urbanization level on the consumption level of rural residents are significantly positive, while the indirect effect is not significant. The result demonstrates that the improvement of urbanization level encourages more farmers to work in cities, which not only improves their consumption level due to the income effect but also helps increase their consumption through changing their cognition, habit, psychology and mode of consumption due to the function of accumulation effect and demonstration effect [32]. However, the level of urbanization has no spatial spillover effect with little influence on the consumption level of rural residents in the adjacent areas.

## 4.6. Regional heterogeneity analysis

Due to the differences in geographical location and factor endowments, there exists obvious heterogeneity in the coupling coordination level of rural consumption system and the consumption level of rural residents in various regions of China. In order to concretely study the regional heterogeneity of the impact of the coupling coordination level of China's rural consumption power system on the consumption level of rural residents, this paper divides 30 provinces into three groups of east, central and west areas. Spatial Dubin time fixed-effect model is constructed respectively for empirical analysis. The results of regional heterogeneity analysis (see Table 8) demonstrate the following findings. (1) The direct effect, indirect effect and total effect of the coupling coordination level of rural consumption power system on the consumption level of rural residents in east area are significantly positive, which indicates that the coupling coordination level of rural consumption power system in east area not only has significant promotional effect on the consumption level of the rural residents in this region but also promotes the consumption level of rural residents in adjacent areas with spatial spillover effect. (2) The direct effect of the coupling coordination level of rural consumption power system on the consumption level of rural residents in central area is positive, while the indirect effect and the total effect are negative, all passing the significance test. It reflects that the coupling coordination level of rural consumption power system obviously promotes the consumption level of rural residents in central area. However, in view of the siphon effect and other reasons, the negative spatial spillover effect suppresses the improvement of the consumption level of the rural residents in the adjacent areas. (3) The direct effect, indirect effect and total effect of the coupling coordination level of the rural consumption system on the consumption level of rural residents in west area are not significant in the test, which indicates that the coupling coordination level of rural consumption power system is low with little effective influence on the improvement of the consumption level of the rural residents in west area. There is no significant spatial spillover effect on the adjacent areas as well.

## 4.7. Robustness test

To ensure the robustness of the empirical results, this paper takes the consumption structure of rural residents as the substitution variable of the explained variable and applies the spatial Durbin model to test the impact of the coupling and coordination of rural social consumption power on the consumption of rural residents. The consumption structure of rural residents is measured by the proportions of development-type consumption and enjoyment-type consumption in residents' living consumption expenditure. The results demonstrate that the total effect coefficient of the coupling and coordination of rural social consumption system on the consumption structure of rural residents is 0.238. The hypothesis test is verified at the significance level of 1%, which indicates that the higher the coupling and coordination level of rural social consumption system in one province is, the more optimized the consumption structure of rural residents is. The spatial lag coefficient of the coupling and coordination level of the rural consumption power system is 0.394 and the null hypothesis is rejected at the significance level of 1%, which indicates that the coupling and coordination development level of the rural consumption

**Table 8. Regional heterogeneity analysis of the influence of coupling coordination level of China's rural consumption power system on the consumption level of rural residents.**

| Variable | Direct effect | Indirect effect | Gross effect |
|---|---|---|---|
| Eastern region | 0.301*** | 0.213* | 0.514** |
| Central region | 0.059*** | −0.154*** | −0.095*** |
| Western region | −0.063 | −0.067 | −0.130 |

Note: In the table, *** , ** and * represent being significant at the level of 1%, 5%, and 10%, respectively.

power system has significant spatial effect. This result shows that the coupling and coordination development level of the rural consumption power system in one province can play a significant role in promoting the optimization of rural residents' consumption structure of its neighbouring provinces. This research further uses the partial differential method to decompose the impact of the coupling coordination level of the rural consumption power system on the consumption structure of rural residents into direct effect, indirect effect and total effect with values of 0.209, 0.129 and 0.338 respectively, which are significant at the 1% level. This result demonstrates that the coupling coordination level of rural consumption power system not only positively impacts the consumption structure of rural residents in a certain province but also induces significant spatial spillover effect. This conclusion is basically in line with the previous estimation results. It can be considered that the empirical results of this research are robust and the conclusions are reliable.

## 5. Research conclusions and policy suggestions

### 5.1. Research conclusions

Based on the theoretical analysis of the mechanism of the coupling and coordination level of rural consumption power systems on the consumption of rural residents, this paper empirically studies the impact of the coupling coordination level of rural consumption power systems on rural residents' consumption through constructing spatial Durbin model based on panel data of 30 Chinese provinces from 2002 to 2022. The research results demonstrate that the coupling coordination level of rural social consumption power system can not only effectively promote the consumption of rural residents in a province but also bring obvious spatial spillover effect, which means that the coupling coordination level of rural social consumption power system in a province has significant promotional effect on the consumption of rural residents in its neighbouring provinces. If the influence of spatial factors is ignored in the process of model construction, the promotional effect of the coupling coordination level of rural social consumption power system on rural residents' consumption will be underestimated. Therefore, it is a must to keep life quality improvement and all-round development of rural residents as the guideline and give full play to the coupling and coordination role of three subsystems of consumption subject, consumption object and consumption environment so as to promote the effective operation of the whole rural social consumption power system and release the huge consumption potential in the rural areas.

### 5.2. Policy suggestions

Along with China's entering into the new development stage of building a modern socialist country in an all-round way, it is impossible to fully activate the consumption kinetic energy of rural residents through simply implementing demand-side management or supply-side structural reform. Supply-side structural reform must be fit with demand-side administration so as to promote the coupling and coordination of the entire rural consumption power system. Therefore, on the one hand, it's important to continue to deepen supply-side structural reform with the goal of enhancing the ability to use supply to guide and create demand. This can be accomplished by improving the adaptability of supply, innovating the supply system, eliminating supply constraints, and enhancing supply capabilities. On the other hand, it's critical to take advantage

of the leading role of demand-side management to constantly rationalize and upgrade the consumption structure through enhancing personal consumption ability, stabilizing consumption expectation, improving consumption quality and releasing consumption potential. By these means, rural residents' consumption can be further promoted and upgraded in both quality and quantity, which can help fundamentally solve the problem of misfit between supply system and rural consumption demand. In fact, the problem of insufficient domestic demand remains the common challenge confronting many economies including India, South Korea and Japan. China's policy experiences and reform trajectories in addressing this issue can be valuable instructions for these nations to formulate proper strategies to stimulate domestic demand and promote consumption.

## Supporting information

**S1 Data. Data.**
(XLS)

**S1 File. Control variables.**
(XLSX)

## Acknowledgments

We would like to thank the School of Business of Yangzhou University, the School of Food and Materials of Nanjing University of Finance and Economics, the Hotel Management colleagues of Macau University of Science and Technology and the School of Economics and Management of Jiangsu University of Science and Technology for their strong support for this work.

## Author contributions

**Methodology:** Rui Wang.

**Resources:** Zhaoqin Chen.

**Writing – original draft:** Zhen Tian.

**Writing – review & editing:** Yan Tan, Zhaoqin Chen.

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
