## [Decision Letter · Decision Letter 0]

3 Sep 2024

PONE-D-24-26863Research on the Impact of Coupling and Coordination of Rural Consumption Power System on the Consumption Level of Rural Residents in ChinaPLOS ONE

Dear Dr. wang,

Thank you for submitting your manuscript to PLOS ONE. After careful consideration, we feel that it has merit but does not fully meet PLOS ONE’s publication criteria as it currently stands. Therefore, we invite you to submit a revised version of the manuscript that addresses the points raised during the review process.

We look forward to receiving your revised manuscript.

Kind regards,

Xingwei Li, Ph.D.

Academic Editor

PLOS ONE

Journal Requirements:

1. When submitting your revision, we need you to address these additional requirements. Please ensure that your manuscript meets PLOS ONE's style requirements, including those for file naming. The PLOS ONE style templates can be found at https://journals.plos.org/plosone/s/file?id=wjVg/PLOSOne_formatting_sample_main_body.pdf and https://journals.plos.org/plosone/s/file?id=ba62/PLOSOne_formatting_sample_title_authors_affiliations.pdf 2. We suggest you thoroughly copyedit your manuscript for language usage, spelling, and grammar. If you do not know anyone who can help you do this, you may wish to consider employing a professional scientific editing service.  The American Journal Experts (AJE) (https://www.aje.com/) is one such service that has extensive experience helping authors meet PLOS guidelines and can provide language editing, translation, manuscript formatting, and figure formatting to ensure your manuscript meets our submission guidelines. Please note that having the manuscript copyedited by AJE or any other editing services does not guarantee selection for peer review or acceptance for publication.  Upon resubmission, please provide the following: The name of the colleague or the details of the professional service that edited your manuscript A copy of your manuscript showing your changes by either highlighting them or using track changes (uploaded as a *supporting information* file) A clean copy of the edited manuscript (uploaded as the new *manuscript* file)” 3. Thank you for stating the following financial disclosure: "This is a part research accomplishment of the project“Research on the Evolution of Rural Residents' Consumption Behavior and the Coordinated Policy of Supply and Demand in the Past Forty Years of Reform" (No. 18BJL004))”, which is supported by Office of Chinese Philosophy and Social Sciences". Please state what role the funders took in the study.  If the funders had no role, please state: "The funders had no role in study design, data collection and analysis, decision to publish, or preparation of the manuscript." If this statement is not correct you must amend it as needed. Please include this amended Role of Funder statement in your cover letter; we will change the online submission form on your behalf. 4. We note that your Data Availability Statement is currently as follows: "All relevant data are within the manuscript and its Supporting Information files." Please confirm at this time whether or not your submission contains all raw data required to replicate the results of your study. Authors must share the “minimal data set” for their submission. PLOS defines the minimal data set to consist of the data required to replicate all study findings reported in the article, as well as related metadata and methods (https://journals.plos.org/plosone/s/data-availability#loc-minimal-data-set-definition). For example, authors should submit the following data: - The values behind the means, standard deviations and other measures reported;- The values used to build graphs;- The points extracted from images for analysis. Authors do not need to submit their entire data set if only a portion of the data was used in the reported study. If your submission does not contain these data, please either upload them as Supporting Information files or deposit them to a stable, public repository and provide us with the relevant URLs, DOIs, or accession numbers. For a list of recommended repositories, please see https://journals.plos.org/plosone/s/recommended-repositories. If there are ethical or legal restrictions on sharing a de-identified data set, please explain them in detail (e.g., data contain potentially sensitive information, data are owned by a third-party organization, etc.) and who has imposed them (e.g., an ethics committee). Please also provide contact information for a data access committee, ethics committee, or other institutional body to which data requests may be sent. If data are owned by a third party, please indicate how others may request data access. 5. When completing the data availability statement of the submission form, you indicated that you will make your data available on acceptance. We strongly recommend all authors decide on a data sharing plan before acceptance, as the process can be lengthy and hold up publication timelines. Please note that, though access restrictions are acceptable now, your entire data will need to be made freely accessible if your manuscript is accepted for publication. This policy applies to all data except where public deposition would breach compliance with the protocol approved by your research ethics board. If you are unable to adhere to our open data policy, please kindly revise your statement to explain your reasoning and we will seek the editor's input on an exemption. Please be assured that, once you have provided your new statement, the assessment of your exemption will not hold up the peer review process. 6. Please amend the manuscript submission data (via Edit Submission) to include authors Dr. Chen Zhao qin and Yan Tan.

Reviewers' comments:

Reviewer's Responses to Questions

**Comments to the Author**

1. Is the manuscript technically sound, and do the data support the conclusions?

Reviewer #1: Yes

Reviewer #2: Partly

2. Has the statistical analysis been performed appropriately and rigorously? 

Reviewer #1: Yes

Reviewer #2: Yes

3. Have the authors made all data underlying the findings in their manuscript fully available?

Reviewer #1: Yes

Reviewer #2: Yes

4. Is the manuscript presented in an intelligible fashion and written in standard English?

Reviewer #1: Yes

Reviewer #2: Yes

5. Review Comments to the Author

Reviewer #1: (1) Abstract: The abstract part is suggested to re-describe. It is necessary to emphasize the innovation, importance and outstanding research results of the research, and select several key data to explain when necessary. In addition, it is recommended to emphasize the significance of the study in the end.

(2) Introduction: The introduction lacks an introduction to the international context. The introduction does not emphasize the innovativeness of the article. The introduction needs to add the study of the former to prove the feasibility of the study. In addition, the last part of the introduction suggests adding explanations to the rest of the article.

(3) Theoretical Analysis and Research hypothesis: This part needs a lot of literature support, it is suggested that the author increase the literature.

(4) A comprehensive evaluation of the coupling coordination level of China's rural consumption power system The specific reasons for the selection of indicators need further explanation

(5) The research results of the article all need to be further explained, not just a description of the results.

(6) Research conclusions and implications: This part is proposed to be changed into conclusions and suggestions. The conclusion and suggestion are divided into two parts.

(7) The article needs to be included in the discussion section.

(8) The article is also in Chinese. I think the author is not serious and needs to be further improved

Reviewer #2: In my personal opinion, the topic of rural consumption power and the consumption level of rural residents is significant and deserving of further exploration. This manuscript investigates the direct impact and spatial spillover effects of rural consumption power on rural residents' consumption in China. However, I believe that this manuscript requires substantial revision. Here are my detailed comments:

1. Unlike other related qualitative studies that primarily focus on rural consumption power and the consumption level of rural residents, this manuscript contributes to the literature by constructing and measuring a comprehensive index of the coupling coordination degree of the rural social consumption power system (D) and examining the indirect effects through a spatial spillover model. However, as the research foundation, the literature review and theoretical analysis parts have not been well elaborated and the author should clarify the logical relationship between rural consumption power and rural residents' consumption and discuss the potential insights we can gain from it.

2. The indicator selection needs to be carefully considered. In this paper, the proxy indicators of the core explanatory variable indicator system and the control variables are reused, such as INC, EDU, FIN, URB, etc. This approach may lead to correlation and bring estimation bias, making it difficult to accurately assess the impact effect, and the author should consider using different indicators or methods to avoid this problem.

3. In Table7, the representation of control variables has changed, and variables such as HEA, CPI, and FSA that have not been described and explained in the previous text have appeared. The author should reconfirm the setting of variables to ensure the credibility of the research conclusion.

4. Another primary concern lies with the references. Since most of the references are from Chinese literature written in Mandarin, it may be challenging for international readers to refer to them. Additionally, relying mostly on studies conducted in China may limit the generalizability of the findings.

5. Some key references appear to be omitted. I noticed that some studies of well-known authors are mentioned in this paper but not cited, such as Adam Smith, Hermann Haken, and others. Additionally, the citation order of references is disordered and not consistent with the numbers in the article. It is recommended that the author recheck the reference format and citation paradigm.

6. We note that the research data is from 2002 to 2019, and based on the data sources in the article, we believe that the author can update the study data to the most recent year.

7. In order to verify the reliability of the research results, the author should adopt various methods to conduct robustness tests, such as replacing the explained variable with survival or development consumption and replacing the core explanatory variable with consumption power lag term.

In terms of minor suggestions:

8. On page 2 and page 22, the phrase "Therefore, neither 'demand-side management' based solely on consumption subjects nor 'supply-side structural reform' based on consumption objects and consumption environments can……in China." appears to overstate the research results of the manuscript, as it does not clearly demonstrate the influence of different subsystems on the consumption level of rural residents.

6. PLOS authors have the option to publish the peer review history of their article (what does this mean? ). If published, this will include your full peer review and any attached files.

**Do you want your identity to be public for this peer review?** For information about this choice, including consent withdrawal, please see our Privacy Policy .

Reviewer #1: No

Reviewer #2: No

---

## [Decision Letter · Decision Letter 1]

28 Jan 2025

PONE-D-24-26863R1Research on the Impact of Coupling and Coordination of Rural Consumption Power System on the Consumption of Rural Residents in ChinaPLOS ONE

Dear Dr. wang,

Thank you for submitting your manuscript to PLOS ONE. After careful consideration, we feel that it has merit but does not fully meet PLOS ONE’s publication criteria as it currently stands. Therefore, we invite you to submit a revised version of the manuscript that addresses the points raised during the review process.

We look forward to receiving your revised manuscript.

Kind regards,

Xingwei Li, Ph.D.

Academic Editor

PLOS ONE

Reviewers' comments:

Reviewer's Responses to Questions

**Comments to the Author**

1. If the authors have adequately addressed your comments raised in a previous round of review and you feel that this manuscript is now acceptable for publication, you may indicate that here to bypass the “Comments to the Author” section, enter your conflict of interest statement in the “Confidential to Editor” section, and submit your "Accept" recommendation.

Reviewer #2: (No Response)

Reviewer #3: (No Response)

2. Is the manuscript technically sound, and do the data support the conclusions?

Reviewer #2: (No Response)

Reviewer #3: (No Response)

3. Has the statistical analysis been performed appropriately and rigorously? 

Reviewer #2: (No Response)

Reviewer #3: (No Response)

4. Have the authors made all data underlying the findings in their manuscript fully available?

Reviewer #2: (No Response)

Reviewer #3: (No Response)

5. Is the manuscript presented in an intelligible fashion and written in standard English?

Reviewer #2: (No Response)

Reviewer #3: (No Response)

6. Review Comments to the Author

**Reviewer #2: ** (No Response)

**Reviewer #3: ** This paper addresses all of the questions I raised, including theoretical analysis, empirical methods, and data handling. The arguments are well-supported and the revisions are appropriate, so I believe the paper can be accepted.

7. PLOS authors have the option to publish the peer review history of their article (what does this mean? ). If published, this will include your full peer review and any attached files.

**Do you want your identity to be public for this peer review?** For information about this choice, including consent withdrawal, please see our Privacy Policy .

Reviewer #2: No

Reviewer #3: No

---

## [Author Response · Author response to Decision Letter 1]

16 Oct 2024

Dear reviewers,

Re: Manuscript ID: PONE-D-24-26863 &Title: Research on the Impact of Coupling and Coordination of Rural Consumption Power System on the Consumption of Rural Residents in China

Thank you for your comments concerning our manuscript. These comments are all valuable and very helpful for us to revise and improve our paper, which has important guiding significance to us. We have studied comments carefully and made improvements on the paper accordingly, which we hope can meet with your approval. All the revised parts are marked in red in the paper. The main improvements on the paper and the responses to the reviews are as follows:

Responses to the comments of Reviewer #1:

Many thanks for your comments. The following are specific responses to each comment.

1. Abstract: The abstract part is suggested to re-describe. It is necessary to emphasize the innovation, importance and outstanding research results of the research, and select several key data to explain when necessary. In addition, it is recommended to emphasize the significance of the study in the end.

Response: We revised the abstract by emphasizing the innovation, importance and significance of the research.

2. Introduction: The introduction lacks an introduction to the international context. The introduction does not emphasize the innovativeness of the article. The introduction needs to add the study of the former to prove the feasibility of the study. In addition, the last part of the introduction suggests adding explanations to the rest of the article.

Response: We revised the introduction part by adding the analysis of the international context, summarizing the existing research and explaining the structure of the paper.

3. Theoretical Analysis and Research hypothesis: This part needs a lot of literature support, it is suggested that the author increase the literature.

Response: On the basis of re-combing the relevant literature, we improved the theoretical analysis and research hypothesis.

4. A comprehensive evaluation of the coupling coordination level of China's rural consumption power system: The specific reasons for the selection of indicators need further explanation.

Response: We added the specific reasons for the selection of the evaluation index of the coupling coordination level of China's rural consumption power system.

5. The research results of the article all need to be further explained, not just a description of the results.

Response: We added the further explanation of the research results in the conclusion part.

6. Research conclusions and implications: This part is proposed to be changed into conclusions and suggestions. The conclusion and suggestion are divided into two parts.

Response: The last part of the manuscript has been divided into two parts: research conclusions and policy suggestions.

7. The article needs to be included in the discussion section.

Response: We added robustness test analysis to further discuss the reliability of the research results.

8. The article is also in Chinese. I think the author is not serious and needs to be further improved

Response: We revised the manuscript according to the review opinions, and the language expression was carefully improved.

Responses to the comments of Reviewer #2:

Special thanks to you for your favorable comments. The following is the specific response.

1.Unlike other related qualitative studies that primarily focus on rural consumption power and the consumption level of rural residents, this manuscript contributes to the literature by constructing and measuring a comprehensive index of the coupling coordination degree of the rural social consumption power system (D) and examining the indirect effects through a spatial spillover model. However, as the research foundation, the literature review and theoretical analysis parts have not been well elaborated and the author should clarify the logical relationship between rural consumption power and rural residents' consumption and discuss the potential insights we can gain from it.

Response: We revised and improved the theoretical analysis and research hypothesis, and added the analysis of the logical relationship between rural social consumption power and rural residents' consumption.

2. The indicator selection needs to be carefully considered. In this paper, the proxy indicators of the core explanatory variable indicator system and the control variables are reused, such as INC, EDU, FIN, URB, etc. This approach may lead to correlation and bring estimation bias, making it difficult to accurately assess the impact effect, and the author should consider using different indicators or methods to avoid this problem.

Response: In constructing the coupling coordination degree index system of rural social consumption power, 29 evaluation indicators are selected based on the three subsystems of consumption subject, consumption object and consumption environment. Some of these indicators like income level and education level are also important variables that affect the consumption of rural households, and no other suitable indicators can be found to replace them. However, when we use these indicators to calculate the coupling coordination degree of the rural consumption power, we mainly measure the degree of variation of each indicator rather than the absolute value, and all the indicators are dimensionless by standardized methods. Therefore, it will not lead to correlation.

3. In Table 7, the representation of control variables has changed, and variables such as HEA, CPI, and FSA that have not been described and explained in the previous text have appeared. The author should reconfirm the setting of variables to ensure the credibility of the research conclusion.

Response: Due to our negligence, there are some errors in the manuscript. We have carefully proofread and improved the full text.

4. Another primary concern lies with the references. Since most of the references are from Chinese literature written in Mandarin, it may be challenging for international readers to refer to them. Additionally, relying mostly on studies conducted in China may limit the generalizability of the findings.

Response: The theory of consumption power is mainly from Marx's classical works, while there is little research findings in the area of neoclassical consumption function theory. Therefore, this research failed to cite much literature of western scholars. In addition, as China is a developing country, its unbalanced economic and social development has led to a low level of coupling and coordination of rural consumption power systems to a certain extent, which limits the expansion of rural consumption. The case study about China can not only provide the basis for the policy measures to expand China's rural consumption, but also has certain reference value for other developing countries.

5. Some key references appear to be omitted. I noticed that some studies of well-known authors are mentioned in this paper but not cited, such as Adam Smith, Hermann Haken, and others. Additionally, the citation order of references is disordered and not consistent with the numbers in the article. It is recommended that the author recheck the reference format and citation paradigm.

Response: We have improved the quality, quantity and format of the reference

6. We note that the research data is from 2002 to 2019, and based on the data sources in the article, we believe that the author can update the study data to the most recent year.

Response: All the data in the manuscript has been updated to 2022.

7. In order to verify the reliability of the research results, the author should adopt various methods to conduct robustness tests, such as replacing the explained variable with survival or development consumption and replacing the core explanatory variable with consumption power lag term.

Response: We test the robustness of the model by means of replacing the explained variable with the proportion of enjoyment-type and development-type consumption in the total consumption expenditure. The empirical results are basically consistent with the previous estimation results. Therefore, it can be considered that the empirical results are robust and the research conclusions are reliable.

8. On page 2 and page 22, the phrase "Therefore, neither 'demand-side management' based solely on consumption subjects nor 'supply-side structural reform' based on consumption objects and consumption environments can……in China." appears to overstate the research results of the manuscript, as it does not clearly demonstrate the influence of different subsystems on the consumption level of rural residents.

Response: Some of the statements in the manuscript are not accurate enough, and we have modified them according to the comments of the reviewer.

---

## [Decision Letter · Decision Letter 2]

24 Apr 2025

PONE-D-24-26863R2Research on the Impact of Coupling and Coordination of Rural Consumption Power System on the Consumption of Rural Residents in ChinaPLOS ONE

Dear Dr. wang,

Thank you for submitting your manuscript to PLOS ONE. After careful consideration, we feel that it has merit but does not fully meet PLOS ONE’s publication criteria as it currently stands. Therefore, we invite you to submit a revised version of the manuscript that addresses the points raised during the review process.

We look forward to receiving your revised manuscript.

Kind regards,

Taiyi He

Academic Editor

PLOS ONE

Journal Requirements:

Additional Editor Comments:

Please consider the expression of power and check the details within this manuscript.

Reviewers' comments:

Reviewer's Responses to Questions

**Comments to the Author**

1. If the authors have adequately addressed your comments raised in a previous round of review and you feel that this manuscript is now acceptable for publication, you may indicate that here to bypass the “Comments to the Author” section, enter your conflict of interest statement in the “Confidential to Editor” section, and submit your "Accept" recommendation.

Reviewer #2: (No Response)

2. Is the manuscript technically sound, and do the data support the conclusions?

Reviewer #2: (No Response)

3. Has the statistical analysis been performed appropriately and rigorously? 

Reviewer #2: (No Response)

4. Have the authors made all data underlying the findings in their manuscript fully available?

Reviewer #2: (No Response)

5. Is the manuscript presented in an intelligible fashion and written in standard English?

Reviewer #2: (No Response)

6. Review Comments to the Author

Reviewer #2: Compared to the previous version, we have noticed that the author has made significant revisions to this article. However, there are still issues with inconsistent citation formats that require further modification, such as formatting issues with references 12, 18, 19, and 28. After revision, I think this article is acceptable for the publication.

7. PLOS authors have the option to publish the peer review history of their article (what does this mean? ). If published, this will include your full peer review and any attached files.

**Do you want your identity to be public for this peer review?** For information about this choice, including consent withdrawal, please see our Privacy Policy .

Reviewer #2: No

---

## [Author Response · Author response to Decision Letter 2]

25 Feb 2025

Dear reviewers,

Re: Manuscript ID: PONE-D-24-26863 & Title: Research on the Impact of Coupling and Coordination of Rural Consumption Power System on the Consumption of Rural Residents in China

Thank you for your comments concerning our manuscript. These comments are all valuable and very helpful for revising and improving our paper, which has important guiding significance to our researches. We have studied comments carefully and made improvements accordingly, which we hope meet with your approval. All the revised parts are marked in red in the paper. The main corrections in the paper and the responses to the reviews are as follows:

1.My major concern is the language used in the revised parts. There are many places that require serious adjustment in order to effectively present ideas. I recommend that professional proofreading is necessary for this paper. The sentences that are confusing and problematic are as follows:

Response: The manuscript has undergone professional language refinement, with meticulous revisions made to address suboptimal linguistic formulations specifically highlighted by the reviewers. Additionally, the manuscript supplements the promotion value of this study in the final policy recommendations.

2. I found no regression results for the robustness test.

Response: We test the robustness of the model by replacing the explained variables with the proportion of enjoyment and development consumption. The results of the robustness test are detailed on page 24.

3. The author also did not arrange the references in the correct format.

Response: We have laid out all the references in APA format.

---

## [Editor Report · Decision Letter 3]

3 Jul 2025

PONE-D-24-26863R3Research on the Impact of Coupling and Coordination of Rural Consumption Power System on the Consumption of Rural Residents in ChinaPLOS ONE

Dear Dr. wang,

Thank you for submitting your manuscript to PLOS ONE. After careful consideration, we feel that it has merit but does not fully meet PLOS ONE’s publication criteria as it currently stands. Therefore, we invite you to submit a revised version of the manuscript that addresses the points raised during the review process.

Upon our initial review, it appears that the response to Reviewer 2’s comment — “I found no regression results for the robustness test.” — is currently missing or insufficiently addressed.

To ensure a smooth review process, we kindly suggest that you revise your response letter to include a specific reply to this point, and resubmit both the updated response letter and the revised manuscript. Please submit your revised manuscript by Aug 17 2025 11:59PM. If you will need more time than this to complete your revisions, please reply to this message or contact the journal office at plosone@plos.org . Please include the following items when submitting your revised manuscript:

We look forward to receiving your revised manuscript.

Kind regards,

Zheng Zhang

Academic Editor

PLOS ONE

Additional Editor Comments:

The authors have done an excellent job—congratulations! Before publication, the statistical procedures and sensitivity analyses should be further reviewed, and the robustness test section should be supplemented with the corresponding regression results.
---

## [Author Response · Author response to Decision Letter 3]

16 May 2025

Dear reviewers,

Re: Manuscript ID: PONE-D-24-26863 & Title: Research on the Impact of Coupling and Coordination of Rural Consumption Power System on the Consumption of Rural Residents in China

Thank you for your comments concerning our manuscript. These comments are all valuable and very helpful for revising and improving our paper, which has important guiding significance to our researches. We have studied comments carefully and made improvements accordingly, which we hope meet with your approval. All the revised parts are marked in red in the paper. The main corrections in the paper and the responses to the reviews are as follows:

1.My major concern is the language used in the revised parts. There are many places that require serious adjustment in order to effectively present ideas. I recommend that professional proofreading is necessary for this paper. The sentences that are confusing and problematic are as follows:

Response: The manuscript has undergone professional language refinement, with meticulous revisions made to address suboptimal linguistic formulations specifically highlighted by the reviewers. Additionally, the manuscript supplements the promotion value of this study in the final policy recommendations.

2.issues with inconsistent citation formats that require further modification, such as formatting issues with references 12, 18, 19, and 28.

Response: We have reorganized the literature for the paper and listed all references in APA format.

---

## [Editor Report · Decision Letter 4]

11 Jul 2025

Research on the Impact of Coupling and Coordination of Rural Consumption Power System on the Consumption of Rural Residents in China

PONE-D-24-26863R4

Dear Dr. Wang,

We’re pleased to inform you that your manuscript has been judged scientifically suitable for publication and will be formally accepted for publication once it meets all outstanding technical requirements.

Kind regards,

Zheng Zhang

Academic Editor

PLOS ONE

Additional Editor Comments (optional):

Thank you for your patience and careful revisions. I believe that throughout the past rounds of review, you have adequately addressed the reviewers' concerns and improved the quality of the manuscript. Therefore, I recommend accepting the manuscript.
---

## [Author Response · Author response to Decision Letter 4]

4 Jul 2025

Thank you for your comments concerning our manuscript. These comments are all valuable and very helpful for us to revise and improve our paper, which has important guiding significance to us. We have studied comments carefully and made improvements on the paper accordingly, which we hope can meet with your approval. All the revised parts are marked in red in the paper. The main improvements on the paper and the responses to the reviews are as follows:

Q: In order to verify the reliability of the research results, the author should adopt various methods to conduct robustness tests, such as replacing the explained variable with survival or development consumption and replacing the core explanatory variable with consumption power lag term.

Response: We test the robustness of the model by means of replacing the explained variable with the proportion of enjoyment-type and development-type consumption in the total consumption expenditure. The empirical results are basically consistent with the previous estimation results. Therefore, it can be considered that the empirical results are robust and the research conclusions are reliable.

To ensure the robustness of the empirical results, this paper takes the consumption structure of rural residents as the substitution variable of the explained variable and applies the spatial Durbin model to test the impact of the coupling and coordination of rural social consumption power on the consumption of rural residents. The consumption structure of rural residents is measured by the proportions of development-type consumption and enjoyment-type consumption in residents’ living consumption expenditure. The results demonstrate that the total effect coefficient of the coupling and coordination of rural social consumption system on the consumption structure of rural residents is 0.238. The hypothesis test is verified at the significance level of 1%, which indicates that the higher the coupling and coordination level of rural social consumption system in one province is, the more optimized the consumption structure of rural residents is. The spatial lag coefficient of the coupling and coordination level of the rural consumption power system is 0.394 and the null hypothesis is rejected at the significance level of 1%, which indicates that the coupling and coordination development level of the rural consumption power system has significant spatial effect. This result shows that the coupling and coordination development level of the rural consumption power system in one province can play a significant role in promoting the optimization of rural residents’ consumption structure of its neighbouring provinces. This research further uses the partial differential method to decompose the impact of the coupling coordination level of the rural consumption power system on the consumption structure of rural residents into direct effect , indirect effect and total effect with values of 0.209, 0.129 and 0.338 respectively, which are significant at the 1% level. This result demonstrates that the coupling coordination level of rural consumption power system not only positively impacts the consumption structure of rural residents in a certain province but also induces significant spatial spillover effect. This conclusion is basically in line with the previous estimation results. It can be considered that the empirical results of this research are robust and the conclusions are reliable.

---

## [Editor Report · Acceptance letter]

PONE-D-24-26863R4

PLOS ONE

Dear Dr. Wang,

I'm pleased to inform you that your manuscript has been deemed suitable for publication in PLOS ONE. Congratulations! Your manuscript is now being handed over to our production team.

Kind regards,

on behalf of

Dr. Zheng Zhang

Academic Editor

PLOS ONE